# Comparison of efficacy and safety of different anticoagulation regimens in plasma exchange: A systematic review and meta-analysis

**Song Ren** **, Liming Huang, Yi Li, Yunlin Feng**\*

Department of Nephrology and Institute of Nephrology, Sichuan Clinical Research Centre for Kidney Diseases, School of Medicine, Sichuan Academy of Medical Sciences & Sichuan Provincial People's Hospital, University of Electronic Science and Technology of China, Chengdu, China

\* fengyunlin@med.uestc.edu.cn

## Abstract

### Background

Extracorporeal line clotting during plasma exchange (PE) not only delays efficient treatment, but also cause great waste of nursing resources. There is a lack of comprehensive comparison of the efficacy and safety among different anticoagulation regimens in plasma exchange in literature.

### Methods

A systematic search was performed in EMBASE, MEDLINE via PubMed, Cochrane Central Library, and CNKI. Studies that had compared at least two anticoagulation regimens in PE were considered eligible. The anticoagulative efficacy outcome was assessed by the occurrence of extracorporeal circuit clotting. The safety outcome was assessed by the occurrence of bleeding events, post-treatment APTT values, and post-treatment platelets counts. The risk of bias was assessed by the AHRQ tool. Mean differences or standardized mean differences with 95% confidence intervals (CIs) of continuous variables and risk ratios (RRs) with 95% CIs of categorical variables were pooled using a random-effects or a fixed-effects model as appropriate.

### Results

In all, 7 studies with 1638 patients and 10951 sessions of PE treatment were included. Pooled results indicated the anticoagulative efficacy of UFH was better than that of saline flushing, yet did not differ with those of LMWH or RCA. Although the occurrence of bleeding events had no difference among different pairs of anticoagulation regimens, anticoagulation using UFH might lead to longer post-treatment APTT value and lower post-treatment platelet counts. Only one study was judged to have low risk of bias in each of the five domains in the AHRQ tool.

**Data Availability Statement:** All relevant data are within the paper and its Supporting information files.

**Funding:** This research was partly funded by Sichuan Province Science and Technology Support Program (2023YFSY0027). The funders had no role in study design, data collection and analysis, decision to publish, or preparation of the manuscript.

**Competing interests:** The authors have declared that no competing interests exist.

## Conclusions

The current anticoagulation regimens are generally effective and well tolerated in PE; however, the number of included studies was too limited to draw definitive conclusions.

## Introduction

Plasma exchange (PE) is a well-established mode of blood purification that is theoretically able to clear all undesired molecules in the plasma [1]. Its application has been extensive in the treatment of various conditions, including liver failure, kidney diseases, autoimmune disorders, neurological diseases, sepsis, and intoxication [2–5]. Adequate coagulation is a critical prerequisite to ensure the effective implementation of PE therapies, thus presenting an important factor to consider during such treatments. Insufficient anticoagulation may lead to premature failure of treatment and great waste of nursing resources, whereas excessive anticoagulation bears high risk of bleeding.

Current pharmaceutical options for anticoagulation in PE include unfractionated heparin (UFH), low molecular weight heparin (LMWH), regional citrate acid (RCA), nafamostat, bivalirudin, and saline flushing [1]. Interestingly, we found in literature most PE treatments had used RCA, especially in Europe [6, 7]; however, in our own clinical practice, LMWH is the most commonly employed anticoagulation regimen, which has demonstrated satisfactory efficacy and safety outcomes. Notably, there is a lack of comprehensive comparison of the efficacy and safety among different anticoagulation regimens in PE in existing literature and no conclusion has been made about the best anticoagulation regimen.

Therefore, we conducted this systematic review and meta-analysis to evaluate the efficacy and safety of different anticoagulation regimens in PE, identify the potentially best regimen, and provide evidence for future development of relevant operation procedures.

## Materials and methods

### Data sources and searches

We conducted a systematic search on 22$^{nd}$ March, 2023 according to the Preferred Reporting Items for Systematic Review and Meta-Analyses (PRISMA) statement [8] for eligible studies in the following electronic data resources without date restriction: EMBASE, MEDLINE via PubMed, Cochrane Central Library, and China National Knowledge Infrastructure (CNKI). The search terms were medical subject headings and text words relevant to PE and anticoagulation (S1 File). This study has been registered on PROSPERO (Identifier# CRD42023413640).

### Study selection

Studies that had compared the outcomes of at least two anticoagulation regimens in PE were considered eligible for inclusion. Based on the preliminary screening experience, eligible studies were restricted to publications after 1990.

Two reviewers (R.S. and H.L.M.) independently conducted the review following a standardized approach. Duplications, non-original studies (e.g., reviews, editorial commentaries, protocols, and guidelines), studies published before 1990, case reports, non-human studies, pediatric studies, studies irrelevant to PE, studies on PE yet without reports on anticoagulation agents, and studies in neither English or Chinese were excluded after careful screening of titles and abstracts. Studies that had only used a single anticoagulation regimen or had not reported

detailed information on coagulation outcomes to allow comparisons were also excluded. Reference lists from full text reviewed articles were further manually screened to identify any other relevant studies. Any discrepancy was adjudicated by a third reviewer (F.Y.L.).

### Definitions of outcomes

The efficacy outcome was assessed by the occurrence of extracorporeal circuit clotting. The safety outcome was assessed by the occurrence of bleeding events, post-treatment APTT values, and post-treatment platelets counts.

### Data extraction

Two reviewers (R.S. and H.L.M.) independently extracted and compiled data from included studies after screening following a double-check procedure. Disagreements were resolved by the third reviewer (F.Y.L.). The data extracted included authors, year of publication, geographical origin, study duration, numbers of patients and procedures, indications for PE, treatment parameters, details of anticoagulation regimens, and details of studied outcomes (S2 File). Information about potential sources of significant clinical heterogeneity, such as age and gender composition of participants, was also collected for potential sensitivity analysis. We have extracted all data needed for this analysis; therefore, we did not need to handle missing data in this study.

### Critical appraisal

Since the included studies contained randomized, nonrandomized, and case-control designs, the study quality was independently assessed by two reviewers (R.S. and H.L.M.) based on the Agency for Healthcare Research and Quality (AHRQ) tool [9].

### Data synthesis and analysis

Data synthesis used Review Manager software (Version 5.2; Cochrane, Oxford, UK). Statistical heterogeneity was estimated using $I^2$ statistic [10]. The statistical heterogeneity of pooled outcomes was deemed as low if $I^2 <25\%$, moderate if $I^2$ ranged from 26% to 75%, and high if $I^2 >75\%$ [11]. For continuous outcomes including post-treatment APTT and platelet count, mean differences (MDs) or standardized mean differences (SMDs) with 95% confidence intervals (CIs) between different paired groups were pooled using a random-effects if $I^2 \geq 25\%$ or a fixed-effects model if $I^2 <25\%$. For categorical outcomes including extracorporeal circuit clotting and bleeding events, risk ratios (RRs) with 95% CIs between different paired groups were pooled using a random-effects or a fixed-effects model based on heterogeneity assessment. The statistical significance was set at a two-sided $p < 0.05$. Funnel plot analysis for publication bias or sensitivity analysis were not performed due to the limited number of studies.

## Results

### Literature searching

5412 records were returned from literature searching after removing duplications. 5305 records excluded after title and abstract screening, leaving 107 records for full text review. After further excluding 100 studies due to having reported only one anticoagulation regimen or lacking sufficient information to allow comparison, seven studies were finally included in this systematic review (Fig 1 and S3 File).

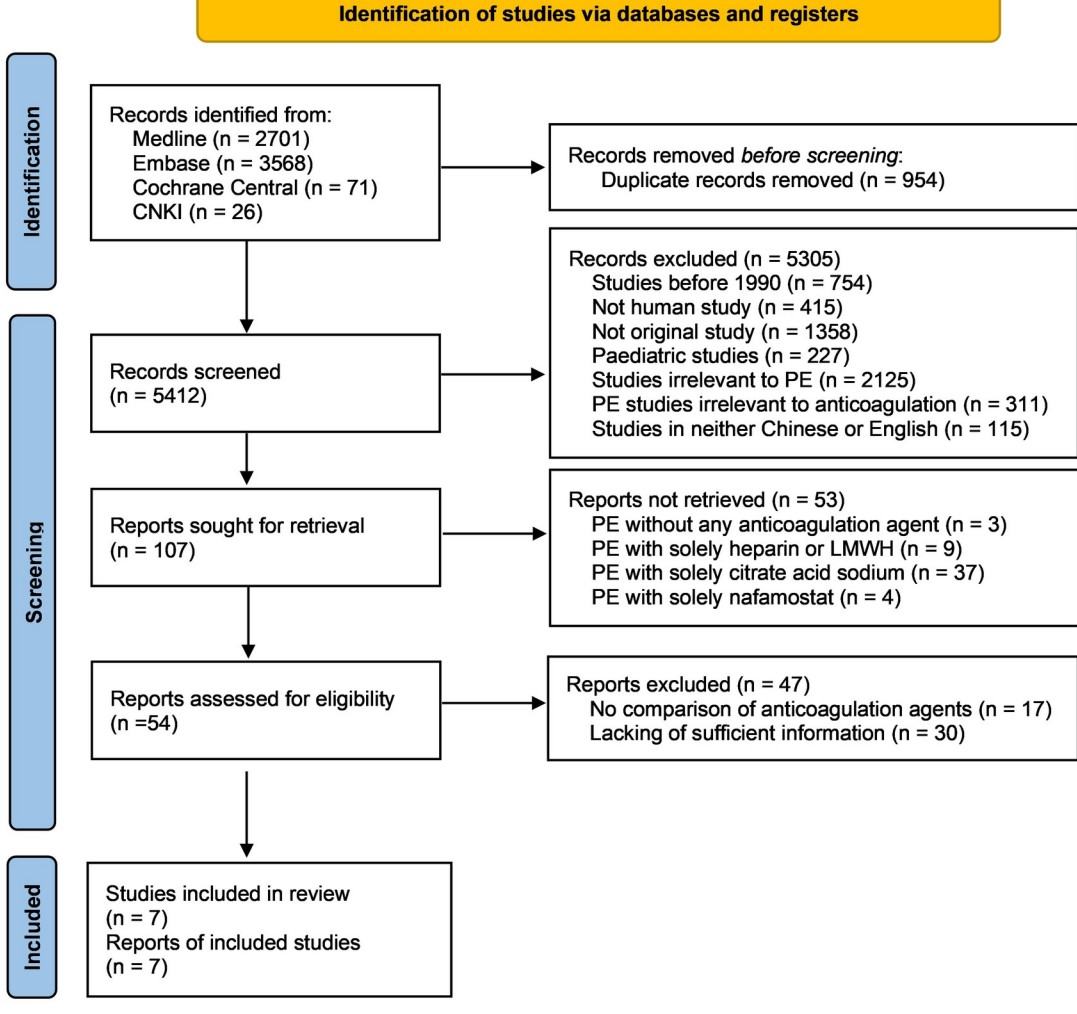

**Fig 1. PRISMA flow chart of this systematic review.**

## Study characteristics

In all, the seven studies involved 1638 patients and 10951 sessions of PE treatment (Table 1). Among these studies, five were respective observational case-control studies [12–16], one was a prospective nonrandomized trial [17], and one was a prospective randomized controlled trial [18]. All studies used dialysis machine and membrane dialyzer to deliver PE treatment. The most common indication for PE was liver failure. Fresh frozen plasma was the most utilized replacement fluid. Two studies had compared three anticoagulation regimens [12, 15], whereas the other five studies had compared two anticoagulation regimens [13, 14, 16–18] (Table 2). All seven studies had reported the outcomes of UFH. There were three studies each that had reported the outcomes of LMWH, RCA, and saline flushing, respectively. Outcomes of extra-corporeal circuit clotting, bleeding, post-treatment APTT values, and post-treatment platelet counts had been reported in six [12–17], seven [12–18], five [13, 15–18], and four [13, 15, 16, 18] studies, respectively (S4 File).

**Table 1. Characteristics of included studies.**

| Author/ Year | Region | Study Design | Study duration | Population, n | Age | Male, n | Targeted diseases | PE Parameters | | | |
|---|---|---|---|---|---|---|---|---|---|---|---|
| | | | | | | | | BFV (ml/ min) | Separation speed (ml/ min) | Replacement fluid | Replacement fluid speed (ml/min) |
| Brunetta, 2017 [12] | Croatia | Respective observation | 1982 to 2014 | 1140 | NR | 476 | 60 conditions, including MG, TMA, SLE, GBS, MS, RPGN, intoxications, etc. | 50–100 | 20–30 | 5% albumin either alone or combined with Ringer's solution or saline; FFP | 20–30 |
| Yuan, 2018 [18] | China | Prospective randomized trial | 2012 to 2014 | 164 | Median: 45 | 148 | Liver failure | 120–130 | 20–40 | FFP | NR |
| Yuan, 2020[15] | China | Respective observation | 2016 to 2017 | 85 | Mean: 54.0 | 50 | Autoimmune disease, liver dysfunction, renal transplantation | 150 | 20 | FFP | NR |
| Teh S, 2022 [14] | Singapore | Retrospective cohort study | 2018 to 2021 | 23 | NR | NR | Kidney transplant recipients | 120–250 | NR | 5% albumin; FFP or cryoprecipitate when needed | NR |
| Ma, 2019 [17] | China | Prospective nonrandomized controlled trial | July to August, 2017 | 52 | NR | 41 | HBV-ACLF | 130 | 30 | FFP | 30 |
| Zhang, 2022 [16] | China | Respective observation | 2020.09 to 2021.03 | 62 | Mean: 50.0 | 42 | Liver failure | 80–110 | 20–50 | FFP | NR |
| Pan, 2015 [13] | China | Respective observation | 2004.04 to 2014.01 | 112 | Mean: 39.0 | 63 | Liver Failure | 80–180 | 20–35 | FFP | NR |

Abbreviations: ACLF, acute-on-chronic liver failure; BFV, blood flow velocity; FFP, fresh frozen plasma; GBS, Gillian-Barre syndrome; MG, myasthenia gravis; min, minutes; ml, milliliter; MS, multiple sclerosis; n, number; NR, not reported; PE, plasma exchange; RPGN, rapidly progressive glomerulonephritis; SD, standard deviation; SLE, systemic lupus erythematosus; TMA, thrombotic microangiopathy.

## Comparison of anticoagulative efficacy

Pooled results of the occurrence of extracorporeal circuit clotting indicated the anticoagulative efficacy of UFH was better than that of saline flushing (RR: 0.33, 95% CI: 0.21 to 0.51, p<0.0001; heterogeneity: $I^2 = 36\%$, p = 0.21), yet did not differ with those of LMWH (RR: 2.73, 95% CI: 0.14 to 54.31, p = 0.51; heterogeneity: $I^2 = 95\%$, p<0.00001) or RCA (RR: 1.28, 95% CI: 0.31 to 5.22, p = 0.73; heterogeneity: $I^2 = 71\%$, p = 0.03) (Fig 2).

## Comparison of safety

The occurrence of bleeding events of UFH did not differ with those of RVA (RR: 2.14, 95% CI: 0.19 to 24.64, p = 0.54; heterogeneity: $I^2 = 52\%$, p = 0.12), saline flushing (RR: 2.09, 95% CI: 0.68 to 6.42, p = 0.20; heterogeneity: $I^2 = 43\%$, p = 0.17), or LMWH (RR: 4.30, 95% CI: 0.10 to 192.47, p = 0.45; heterogeneity: $I^2 = 90\%$, p<0.0001) (Fig 3). Pooled results indicated the post-treatment APTT value of UFH was consistently longer than those of RCA (SMD: 1.51s, 95% CI: 1.09s to 1.93s, p<0.001; heterogeneity: $I^2 = 0\%$, p = 0.62), saline flushing (SMD: 1.42s, 95% CI: 0.99s to 1.85s, p<0.001; heterogeneity: $I^2 = 48\%$, p = 0.17), and LMWH (SMD: 0.40s, 95% CI: 0.19s to 0.61s, p<0.001; heterogeneity: $I^2 = 0\%$, p = 0.80) (Fig 4). The pooled post-treatment platelet count of UFH was significantly less than that of LMWH (MD: -25.45x$10^9$/L, 95% CI: -30.83x$10^9$/L to -20.07x$10^9$/L, p<0.001; heterogeneity: $I^2 = 0\%$, p = 0.66), yet did not differ

**Table 2. Anticoagulation regimens and outcomes of included studies.**

| Author/Year | Procedures, n | UFH | | LMWH | | RCA | | Saline flushing |
|---|---|---|---|---|---|---|---|---|
| | | n* | Protocol | n* | Protocol | n* | Protocol | n* |
| Brunetta, 2017 [12] | 9611 | 7733 | 50 IU/kg+1000 IU/h | 575 | nadroparin: 65 IU/kg; enoxaparin: 100 IU/kg; daltaparin: 65 IU/kg; reviparin: 50 IU/kg | - | | 1193 |
| Yuan, 2018 [18] | 398 | 168 | 2500 IU+50 IU/h | - | - | - | - | 230 |
| Yuan, 2020 [15] | 255 | 120 | 40 IU/kg+625–1000 IU/h | - | - | 93 | 170ml/h, adjusted to match post-filter iCa of 0.25–0.45 mmol/L | 42 |
| Teh S, 2022 [14] | 112 | 50 | 2000 IU+1000 IU/h or 500–1000 IU+250–500 IU/h | - | - | 62 | 120–150 ml/h, adjusted to match post-filter iCa of 0.25–0.35 mmol/L | - |
| Ma, 2019 [17] | 120 | 94 | 3125 IU+500 IU/h | - | - | 106 | 100 ml/h | - |
| Zhang, 2022 [16] | 83 | 62 | NR | 21 | NR | - | - | - |
| Pan, 2015 [13] | 372 | 108 | NR | 264 | NR | - | - | - |

* number of procedures.

Note: "-" represents treatment regimens that were not included in the study.

Abbreviations: iCa, ionized calcium; kg, kilogram; L, liter; LMWH, low molecular weight heparin; n, number; NR, not reported; RCA, regional citrate acid; UFH, unfractionated heparin.

with that of saline flushing (MD: -1.66x10$^9$/L, 95% CI: -6.30x10$^9$/L to 2.99x10$^9$/L, p<0.001; heterogeneity: I$^2$ = 0%, p = 0.68) (Fig 5).

## Critical appraisal

Only one study was judged to have low risk of bias in each of the five domains in the AHRQ tool [17], and four studies had domains with high risk of bias (Fig 6). The two domains with the highest proportions of high risk of bias were attrition bias and reporting bias (both 2/7, 28.6%, see in S5 File).

## Discussion

Pooled results of comparisons between different pairs of anticoagulation regimens in PE indicated the anticoagulative efficacy of each anticoagulation regimen did not differ among each other, yet consistently better than that of saline flushing. Although the occurrence of bleeding events had no difference, anticoagulation using UFH might lead to longer post-treatment APTT value and lower post-treatment platelet counts. Critical appraisal showed more than half of the studies had high risk of bias based on the AHRQ assessment. It should be noted that the number of included studies was too limited to draw definitive conclusion on the best anticoagulation regimen in PE.

Anticoagulative drug is not the only determinant of anticoagulative efficacy in PE, which is influenced by multiple other factors including but not limited to filter membrane, blood flow rate, plasma separation speed, and replacement fluid speed [19]. It should bear in mind when interpret the findings of this study, the limited number of included studies precluded comparisons of anticoagulation regimens with the above cofounders adjusted. The choice of

(A) Comparison of the effects of UFH and RCA on extracorporeal circuit clotting

(B) Comparison of the effects of UFH and saline flushing on extracorporeal circuit clotting

(C) Comparison of the effects of UFH and LMWH on extracorporeal circuit clotting

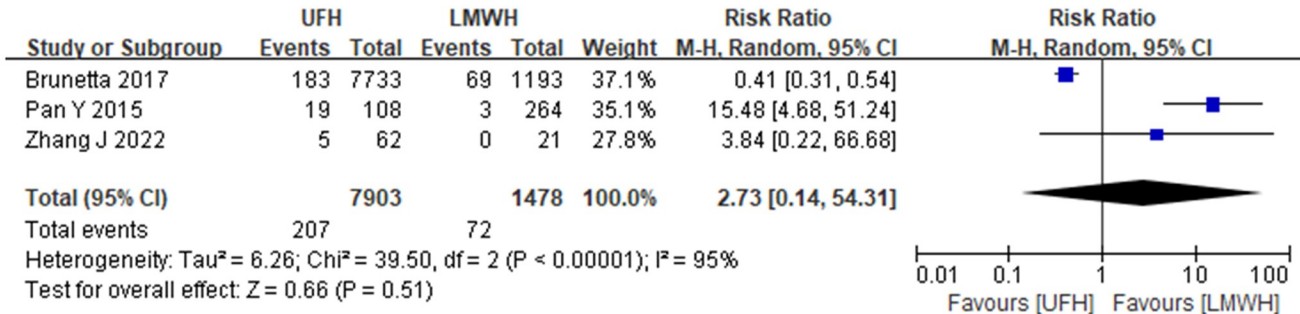

**Fig 2. Comparisons of the effects of different anticoagulation regimens on extracorporeal circuit clotting.**

anticoagulation regimen is also influenced by the indications of PE, experience and preference of practitioners, and local operation procedures. For example, although RCA has been reported safe in patients with liver diseases [17, 20], we usually use UFH or LMWH in PE for liver failure patients in our local practice. Saline flushing is also used in patients with low platelet counts or coagulative disorders. In addition, the most commonly reported indication for PE in European countries such as Italy is neurological disease, which might partly explain why RCA is most often used [21–23].

Generally, all current anticoagulation regimens are well tolerated. UFH interacts with multiple targets in the coagulative cascade, including Factors IIa, IXa, Xa, XIa, and XIIa [24]. The anticoagulative effects of moderate and high dose of UFH can be monitored by APTT and ACT, respectively [24]. Although the pooled results showed the APTT and PLT values after

(A) Comparison of the effects of UFH and RCA on bleeding events

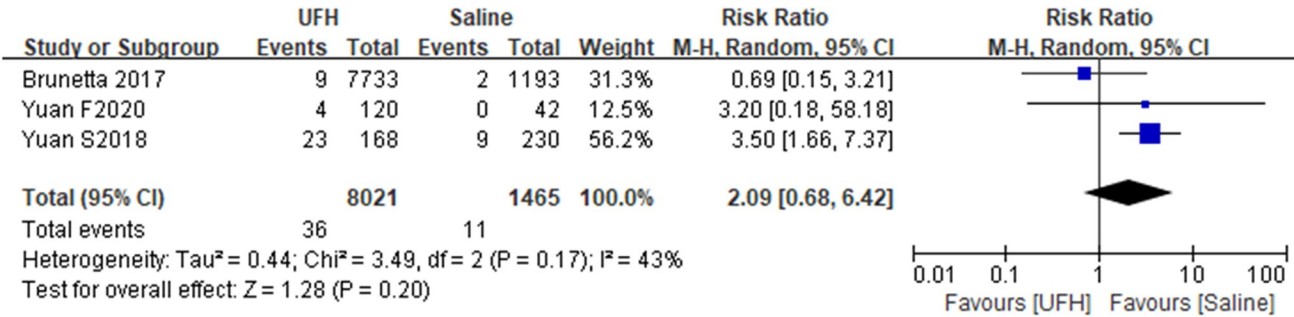

(B) Comparison of the effects of UFH and saline flushing on bleeding events

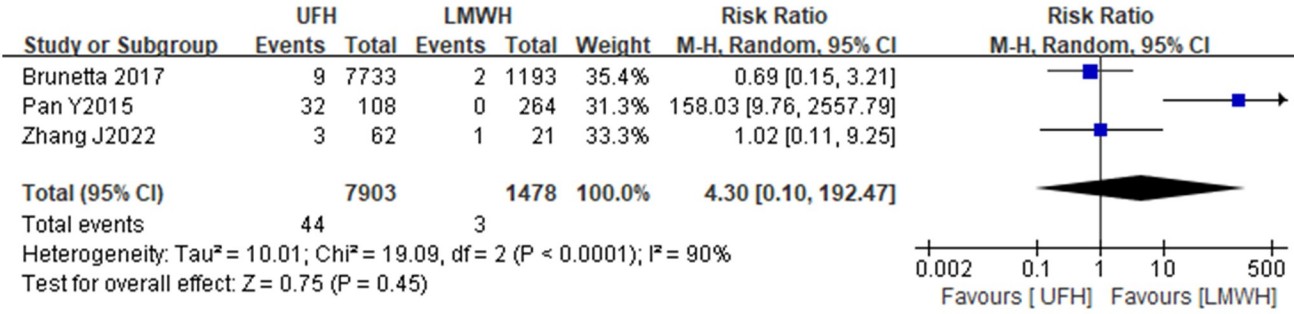

(C) Comparison of the effects of UFH and LMWH on bleeding events

**Fig 3. Comparisons of the effects of different anticoagulation regimens on bleeding events.**

treatment were worse in UFH anticoagulation settings, the occurrence of bleeding events did not differ among all anticoagulation regimens. Therefore, UFH did not exhibited apparent disadvantages in PE; however, its use should be carefully balanced in patients with pre-existing coagulative disorders and/or low PLT counts. These two clinical settings are commonly observed in patients with liver failure or thrombotic microangiopathy, which are both important indications for PE treatment. The growing utilization of novel anticoagulant agents, such as rivaroxaban, has the potential to introduce new clinical scenarios for PE. For instance, patients with nephrotic syndrome who are receiving rivaroxaban may require PE treatment under specific clinical settings, such as during the outbreak of underlying autoimmune diseases. In such instances, rivaroxaban becomes a crucial consideration when prescribing

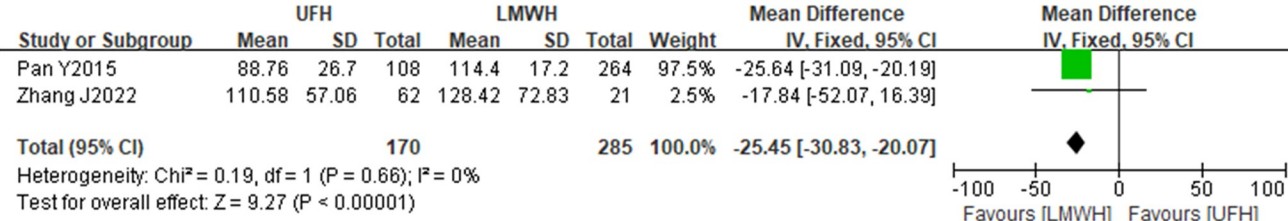

### (A) Comparison of the effects of UFH and RCA on APTT

| Study or Subgroup | UFH Mean | SD | Total | RCA Mean | SD | Total | Weight | Std. Mean Difference IV, Fixed, 95% CI |
|---|---|---|---|---|---|---|---|---|
| Ma Y 2019 | 162.7 | 27.2 | 24 | 122.5 | 29.3 | 28 | 46.3% | 1.40 [0.78, 2.01] |
| Yuan F 2020 | 48.7 | 3.6 | 34 | 40.1 | 6.7 | 30 | 53.7% | 1.61 [1.04, 2.18] |
| **Total (95% CI)** | | | **58** | | | **58** | **100.0%** | **1.51 [1.09, 1.93]** |

Heterogeneity: Chi² = 0.25, df = 1 (P = 0.62); I² = 0%
Test for overall effect: Z = 7.10 (P < 0.00001)

### (B) Comparison of the effects of UFH and saline flushing on APTT

| Study or Subgroup | UFH Mean | SD | Total | Saline Mean | SD | Total | Weight | Std. Mean Difference IV, Random, 95% CI |
|---|---|---|---|---|---|---|---|---|
| Yuan F2020 | 48.7 | 3.6 | 34 | 41 | 5.3 | 21 | 30.1% | 1.76 [1.11, 2.40] |
| Yuan S2018 | 104.1 | 41.3 | 164 | 60.6 | 24.8 | 164 | 69.9% | 1.27 [1.04, 1.51] |
| **Total (95% CI)** | | | **198** | | | **185** | **100.0%** | **1.42 [0.99, 1.85]** |

Heterogeneity: Tau² = 0.06; Chi² = 1.91, df = 1 (P = 0.17); I² = 48%
Test for overall effect: Z = 6.41 (P < 0.00001)

### (C) Comparison of the effects of UFH and LMWH on APTT

| Study or Subgroup | UFH Mean | SD | Total | LMWH Mean | SD | Total | Weight | Std. Mean Difference IV, Fixed, 95% CI |
|---|---|---|---|---|---|---|---|---|
| Pan Y 2015 | 71.67 | 22.47 | 108 | 62.15 | 23.32 | 264 | 82.9% | 0.41 [0.19, 0.64] |
| Zhang J 2022 | 61.37 | 38.74 | 62 | 49.38 | 16.93 | 21 | 17.1% | 0.34 [-0.15, 0.84] |
| **Total (95% CI)** | | | **170** | | | **285** | **100.0%** | **0.40 [0.19, 0.61]** |

Heterogeneity: Chi² = 0.06, df = 1 (P = 0.80); I² = 0%
Test for overall effect: Z = 3.81 (P = 0.0001)

**Fig 4. Comparisons of the effects of different anticoagulation regimens on post-treatment APTT.**

### (A) Comparison of the effects of UFH and saline flushing on platelet counts

| Study or Subgroup | UFH Mean | SD | Total | Saline Mean | SD | Total | Weight | Mean Difference IV, Fixed, 95% CI |
|---|---|---|---|---|---|---|---|---|
| Yuan F2020 | 98 | 11.2 | 34 | 99 | 9.7 | 21 | 68.7% | -1.00 [-6.60, 4.60] |
| Yuan S2018 | 81.8 | 45.7 | 164 | 84.9 | 29.2 | 164 | 31.3% | -3.10 [-11.40, 5.20] |
| **Total (95% CI)** | | | **198** | | | **185** | **100.0%** | **-1.66 [-6.30, 2.99]** |

Heterogeneity: Chi² = 0.17, df = 1 (P = 0.68); I² = 0%
Test for overall effect: Z = 0.70 (P = 0.48)

### (B) Comparison of the effects of UFH and LMWH on platelet counts

| Study or Subgroup | UFH Mean | SD | Total | LMWH Mean | SD | Total | Weight | Mean Difference IV, Fixed, 95% CI |
|---|---|---|---|---|---|---|---|---|
| Pan Y2015 | 88.76 | 26.7 | 108 | 114.4 | 17.2 | 264 | 97.5% | -25.64 [-31.09, -20.19] |
| Zhang J2022 | 110.58 | 57.06 | 62 | 128.42 | 72.83 | 21 | 2.5% | -17.84 [-52.07, 16.39] |
| **Total (95% CI)** | | | **170** | | | **285** | **100.0%** | **-25.45 [-30.83, -20.07]** |

Heterogeneity: Chi² = 0.19, df = 1 (P = 0.66); I² = 0%
Test for overall effect: Z = 9.27 (P < 0.00001)

**Fig 5. Comparisons of the effects of different anticoagulation regimens on post-treatment platelet counts.**

## (A) Summary graph of risk of bias

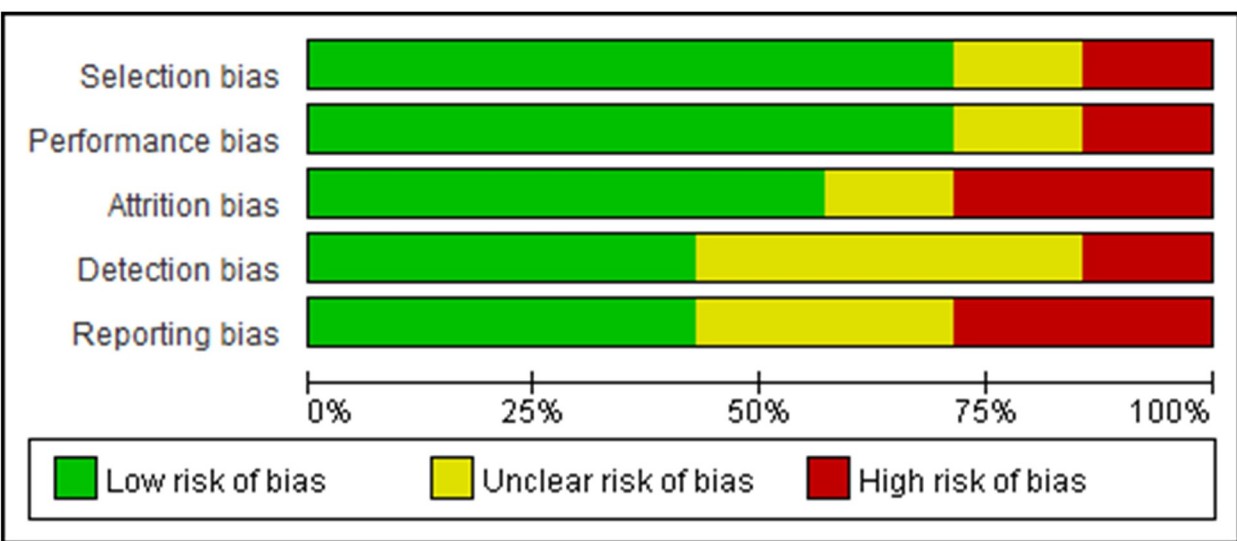

## (B) Traffic light graph of risk of bias

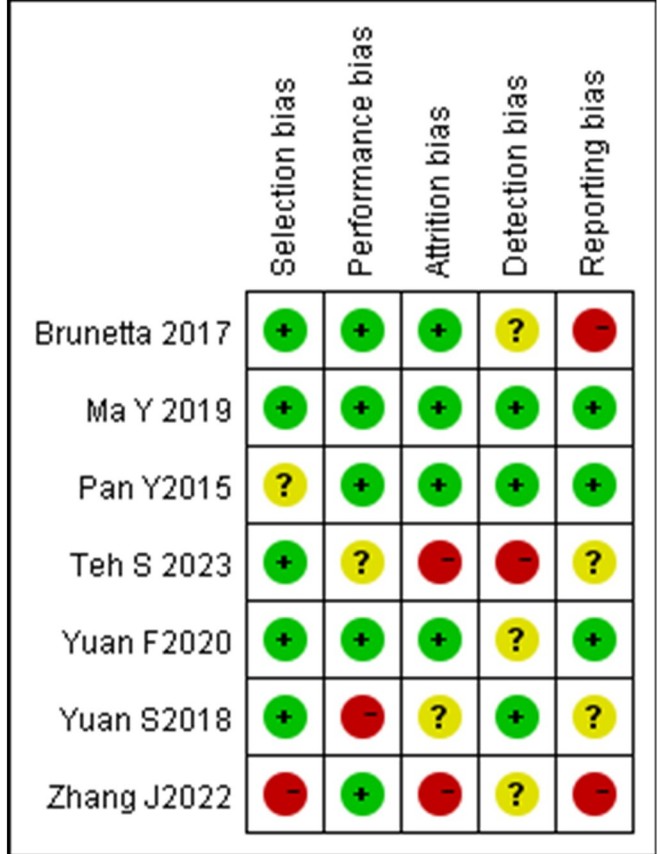

**Fig 6. Quality assessment results of included studies based on the AHRQ tool.**

anticoagulants for PE. Unfortunately, there is a lack of literature addressing this particular application. Future investigations are warranted to provide further insights into this area.

To the best of our acknowledgment, this is the first systematic review and meta-analysis on comparisons of different anticoagulation regimens in PE. Several limitations need to be acknowledged. Firstly, the limited number of studies included in this review precluded the ability to derive definitive conclusions, conduct sensitivity analysis, or analyze publication bias. It was also the reason that network meta-analysis was deemed unfeasible. Secondly, the majority of the included studies focused on liver failure populations, thereby failing to encompass the broader indications for PE. Lastly, the comparisons were unable to account for factors that might have influenced the observed anticoagulative outcomes beyond anticoagulative drugs, such as blood flow. More studies especially well-designed randomized controlled trials (RCTs) are needed for further investigations on the benefits and risks of different anticoagulation regimens in PE.

## Conclusions

The findings of this study indicate the current anticoagulation regimens are generally effective and well-tolerated to ensure successful delivering of PE treatments. Although the occurrence of bleeding events had no difference, UFH anticoagulation might lead to longer post-treatment APTT value and lower post-treatment platelet counts. The number of included studies was too limited to draw definitive conclusion in this field. More studies especially well-designed RCTs are needed to balance the benefits and risks of different anticoagulation regimens in PE.

## Supporting information

**S1 File. Literature search strategies.**
(DOCX)

**S2 File. The data extracted from the studies included in this systematic review that would be needed to replicate this meta-analysis.**
(DOCX)

**S3 File. Detailed information of excluded studies.**
(DOCX)

**S4 File. Reported outcomes of included studies.**
(DOCX)

**S5 File. The bias risk for each study in this meta-analysis based on the Cochrane tool.**
(DOCX)

## Acknowledgments

We sincerely thank Mr. Qiang Li for the help in data analysis.

## Author Contributions

**Conceptualization:** Song Ren, Liming Huang, Yunlin Feng.

**Data curation:** Song Ren, Liming Huang, Yunlin Feng.

**Formal analysis:** Yi Li.

**Methodology:** Song Ren, Yi Li.

**Software:** Song Ren, Liming Huang, Yunlin Feng.

**Validation:** Yi Li.

**Writing – original draft:** Song Ren, Yi Li.

**Writing – review & editing:** Yunlin Feng.

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
