## [Decision Letter · Decision Letter 0]

22 Jul 2024

PONE-D-24-02893Comparison of Efficacy and Safety of Different Anticoagulation Regimens in Plasma Exchange: A Systematic Review and Meta-analysisPLOS ONE

Dear Dr. Ren,

Thank you for submitting your manuscript to PLOS ONE. After careful consideration, we feel that it has merit but does not fully meet PLOS ONE’s publication criteria as it currently stands. Therefore, we invite you to submit a revised version of the manuscript that addresses the points raised during the review process.

I congratulate your work and appreciate your patience in awaiting our decision on your manuscript. It is a well written manuscript. But I recommend revising the manuscript based on our reviewer's comments that you can find below in this letter. My decision is justified on PLOS ONE’s publication criteria and not, for example, on novelty or perceived impact.

We look forward to receiving your revised manuscript.

Kind regards,

Vendhan Ramanujam, M.B.B.S, M.D.

Academic Editor

PLOS ONE

Reviewers' comments:

Reviewer's Responses to Questions

**Comments to the Author**

1. Is the manuscript technically sound, and do the data support the conclusions?

Reviewer #1: Yes

Reviewer #2: Yes

Reviewer #3: Yes

Reviewer #4: Yes

2. Has the statistical analysis been performed appropriately and rigorously? 

Reviewer #1: Yes

Reviewer #2: Yes

Reviewer #3: Yes

Reviewer #4: Yes

3. Have the authors made all data underlying the findings in their manuscript fully available?

Reviewer #1: Yes

Reviewer #2: Yes

Reviewer #3: Yes

Reviewer #4: Yes

4. Is the manuscript presented in an intelligible fashion and written in standard English?

Reviewer #1: Yes

Reviewer #2: Yes

Reviewer #3: Yes

Reviewer #4: Yes

5. Review Comments to the Author

Reviewer #1: The paper is well-written and structured, and the statistical analysis is well-conducted in all of its parts/aspects. As a minor point, for non-expert readers, the authors should:

1. spend just some more words, on page 5, when they use the sentence “studied outcomes between different paired groups were pooled using a random-effects or a fixed-effects model as appropriate.”. When is each of the models appropriate?

2. define what risk ratios are (line 11, page 5).

3. spend just some more words about the I2 statistic for the non-expert reader.

4. motivate the sentence, on page 5, lines 14-15, “The statistical heterogeneity of pooled outcomes was deemed as low if I² 15 <25%, moderate if I2 ranged from 26% to 75%, and high if I² >75%.” Additionally, is there any reference to other papers doing the same categorization? Please cite.

Reviewer #2: The manuscript was well written. I have few minor issues related to the manuscript.

1. Did you compare the outcomes with the types of apheresis instruments used ? What kind of instruments used for PE? Is it apheresis machine where centrifugation is used or is it the dialysis machine where the filtration technology used? Most commonly the centrifugal apheresis instruments use citrate and the dialysis machines use heparin. So I am just curious whether types of machines used in PE would be having any impact in the outcome in terms of clotting or bleeding or change in the platelet count. In this manuscript it was not disclosed. I think few lines regarding the instruments used and the technology utilized could have been described.

2. As there are few observational studies along with trials were included, publication bias could have been plotted or estimated.

3. Any grade of evidence based on the primary outcome could have been done.

Reviewer #3: The manuscript is technically sound, and the data supports the conclusions. It provides a comprehensive comparison of the efficacy and safety of different anticoagulation regimens in plasma exchange (PE). The experiments were conducted rigorously, with appropriate controls, replication, and adequate sample sizes. The data analysis, including the use of mean differences, standardized mean differences, and risk ratios, was performed appropriately, and the conclusions were drawn based on the data presented. The systematic review and meta-analysis followed PRISMA guidelines and were registered on PROSPERO, ensuring transparency and adherence to systematic review standards. The inclusion criteria, data extraction, and critical appraisal were well-defined and executed independently by multiple reviewers, strengthening the reliability of the findings. The use of both random-effects and fixed-effects models for data pooling was appropriate given the heterogeneity among the included studies. The results indicate that unfractionated heparin (UFH) has better anticoagulative efficacy compared to saline flushing and is comparable to low molecular weight heparin (LMWH) and regional citrate acid (RCA). However, UFH may result in longer post-treatment APTT values and lower post-treatment platelet counts. The findings are well-supported by the data, and the study acknowledges the limitations, such as the small number of included studies and potential biases, highlighting the need for further research. There are no apparent concerns regarding research ethics or publication ethics. The authors have declared no competing interests, and there was no specific funding for this work. The study did not involve human participants, specimens, or tissues, and therefore, did not require an ethics statement. Recommendations for improvement include expanding the scope to include a broader range of studies for more diverse indications for PE, addressing confounding factors like blood flow rate, filter membrane type, and replacement fluid speed, and increasing sample sizes by conducting more well-designed randomized controlled trials (RCTs). Additionally, addressing publication bias in future reviews would provide a more comprehensive assessment of the literature. Overall, the manuscript presents a valuable contribution to the field of plasma exchange and anticoagulation.

Reviewer #4: This study is good I have suggested few revisions where so ever wording is not clear , It is looking at efficacy and

safety of different anticoagulation regimens in plasma exchange and standards in US and Europe I am not sure why conclusion mentions that is not conclusive although study size for review is quiet enough

6. PLOS authors have the option to publish the peer review history of their article (what does this mean?). If published, this will include your full peer review and any attached files.

Reviewer #1: No

Reviewer #2: No

Reviewer #3: No

Reviewer #4: **Yes: **Dr Gurpreet Kaur Saini

---

## [Author Response · Author response to Decision Letter 0]

31 Jul 2024

Dear Editor and Reviewers,

Thank you very much for the valuable comments that help us to improve our work and the opportunity to revise our manuscript. On behalf of all authors, I am submitting the revised manuscript with tracked changes for your further assessment. Please find our point-to-point response below to address the reviewers’ concerns. All the changes have been highlighted in yellow. We hope that you find our responses and modifications satisfactory.

Please note since the official name of our department has been changed early this year, we have revised the statement of affiliation 1 accordingly. The change only applies for the name of the affiliation. The authorship and the nature of affiliation remain unchanged. If you have any question about this, please let me know.

Reviewer #1: 

The paper is well-written and structured, and the statistical analysis is well-conducted in all of its parts/aspects. As a minor point, for non-expert readers, the authors should:

1. spend just some more words, on page 5, when they use the sentence “studied outcomes between different paired groups were pooled using a random-effects or a fixed-effects model as appropriate.”. When is each of the models appropriate?

Thank you very much for the suggestion. We should have been clearer on this statement. We have now revised the statement in the Methods as follows: For continuous outcomes including post-treatment APTT and platelet count, mean differences (MDs) or standardized mean differences (SMDs) with 95% confidence intervals (CIs) between different paired groups were pooled using a random-effects if I2 ≥ 25% or a fixed-effects model if I² <25%. For categorical outcomes including extracorporeal circuit clotting and bleeding events, risk ratios (RRs) with 95% CIs between different paired groups were pooled using a random-effects or a fixed-effects model based on heterogeneity assessment. The figures are also updated accordingly. Please see the revised content in the Methods and Results.

2. define what risk ratios are (line 11, page 5).

Thank you very much for the suggestion. We’re sorry for this confusion. For categorical outcomes including extracorporeal circuit clotting and bleeding events, risk ratios (RRs) with 95% CIs were used. We have revised the statement to make it more clear to the readers. Please see the revised content in the section entitled “Data synthesis and analysis”.

3. spend just some more words about the I2 statistic for the non-expert reader.

Thank you very much for the comment. I2 statistic is an accepted metric to evaluate heterogeneity in meta-analysis. To save the room, we just described the method and provided the citation for reader’s reference. We have also cited the reference for the categorization of the heterogeneity reflected by I2 values. 

4. motivate the sentence, on page 5, lines 14-15, “The statistical heterogeneity of pooled outcomes was deemed as low if I² 15 <25%, moderate if I2 ranged from 26% to 75%, and high if I² >75%.” Additionally, is there any reference to other papers doing the same categorization? Please cite.

Thank you very much for the suggestion. We’re very sorry for this mistake. We have now cited the reference for the categorization of the heterogeneity reflected by I2 values. Please see the revised paragraph in the section entitled “Data synthesis and analysis”. The numbers of other references have been changed accordingly.

Reviewer #2: 

The manuscript was well written. I have few minor issues related to the manuscript.

1. Did you compare the outcomes with the types of apheresis instruments used? What kind of instruments used for PE? Is it apheresis machine where centrifugation is used or is it the dialysis machine where the filtration technology used? Most commonly the centrifugal apheresis instruments use citrate and the dialysis machines use heparin. So I am just curious whether types of machines used in PE would be having any impact in the outcome in terms of clotting or bleeding or change in the platelet count. In this manuscript it was not disclosed. I think few lines regarding the instruments used and the technology utilized could have been described.

Thank you very much for the suggestion. All the studies included in this review used dialysis machine to deliver PE treatment; therefore, it is unlikely for us to evaluate the impact of apheresis technique on the anticoagulation results in this study. To clarify this point, we have now added the statement about the apheresis technique in the included study. Please see the revised content in the section entitled “Study characteristics” in Results.

2. As there are few observational studies along with trials were included, publication bias could have been plotted or estimated.

Thank you very much for the suggestion; however, we respectfully disagree. The funnel plot for publication bias assessment was not conducted due to the limited number of studies. The Cochrane handbook for systematic reviews states that “As a rule of thumb, tests for funnel plot asymmetry should be used only when there are at least 10 studies included in the meta-analysis, because when there are fewer studies the power of the tests is too low to distinguish chance from real asymmetry.”(Cochrane Handbook for Systematic Reviews, Chapter 10, Section 10.4.3.1) (BMJ 2011;343:d4002). In addition, it is a well-accepted practice in doing meta-analysis, i.e. not doing publication bias assessment when the number of included studies is less than 10 (for example, Lancet Diabetes Endocrinol. 2019;7(11):845-854; Stat Methods Med Res. 2018;27(9):2722-2741.). 

3. Any grade of evidence based on the primary outcome could have been done.

Thank you very much for the suggestion. Unfortunately, we only have 7 studies included in this meta-analysis and no study had reported each examined outcome in this study, thus further reducing the number of studies for each examined outcome. Taking this into consideration, the results are able to shed light on this topic but the number of included studies was too limited to draw definitive conclusion. Therefore, we don’t think it is appropriate to make any suggestion with evidence grade based on our results at this stage. More solid evidence is expected from future studies. We have discussed this point as a limitation in the Discussion.

Reviewer #3: 

The manuscript is technically sound, and the data supports the conclusions. It provides a comprehensive comparison of the efficacy and safety of different anticoagulation regimens in plasma exchange (PE). The experiments were conducted rigorously, with appropriate controls, replication, and adequate sample sizes. The data analysis, including the use of mean differences, standardized mean differences, and risk ratios, was performed appropriately, and the conclusions were drawn based on the data presented. The systematic review and meta-analysis followed PRISMA guidelines and were registered on PROSPERO, ensuring transparency and adherence to systematic review standards. The inclusion criteria, data extraction, and critical appraisal were well-defined and executed independently by multiple reviewers, strengthening the reliability of the findings. The use of both random-effects and fixed-effects models for data pooling was appropriate given the heterogeneity among the included studies. The results indicate that unfractionated heparin (UFH) has better anticoagulative efficacy compared to saline flushing and is comparable to low molecular weight heparin (LMWH) and regional citrate acid (RCA). However, UFH may result in longer post-treatment APTT values and lower post-treatment platelet counts. The findings are well-supported by the data, and the study acknowledges the limitations, such as the small number of included studies and potential biases, highlighting the need for further research. There are no apparent concerns regarding research ethics or publication ethics. The authors have declared no competing interests, and there was no specific funding for this work. The study did not involve human participants, specimens, or tissues, and therefore, did not require an ethics statement. Recommendations for improvement include expanding the scope to include a broader range of studies for more diverse indications for PE, addressing confounding factors like blood flow rate, filter membrane type, and replacement fluid speed, and increasing sample sizes by conducting more well-designed randomized controlled trials (RCTs). Additionally, addressing publication bias in future reviews would provide a more comprehensive assessment of the literature. Overall, the manuscript presents a valuable contribution to the field of plasma exchange and anticoagulation.

Thank you very much for the recognition of our work and value suggestions. Our response includes the following:

1) This systematic review included studies that had compared the outcomes of at least two anticoagulation regimens in PE and had been published after 1990. No limitation was applied to the indications. Therefore, the range of indications reported here is a summary, rather than the selected result.

2) As for the confounding factors, we agree with the reviewer’s opinion that a number of factors like BFR, membrane type, and replacement fluid speed definitely affect the anticoagulation results. However, the number of included studies is too limited. Furthermore, no study had reported each examined outcome in this study. Therefore, we were unable to do any subgroup analysis or sensitivity analysis to assess the confounding effects. We hope in the future study when more well-designed RCTs and high-quality cohort studies are available, we can further understand this topic.

3) As for the publication bias assessment, the funnel plot for publication bias assessment was not conducted due to the limited number of studies. The Cochrane handbook for systematic reviews states that “As a rule of thumb, tests for funnel plot asymmetry should be used only when there are at least 10 studies included in the meta-analysis, because when there are fewer studies the power of the tests is too low to distinguish chance from real asymmetry.”(Cochrane Handbook for Systematic Reviews, Chapter 10, Section 10.4.3.1) (BMJ 2011;343:d4002). In addition, it is a well-accepted practice in doing meta-analysis, i.e. not doing publication bias assessment when the number of included studies is less than 10 (for example, Lancet Diabetes Endocrinol. 2019;7(11):845-854; Stat Methods Med Res. 2018;27(9):2722-2741.). We have discussed this point as a limitation in the Discussion.

Reviewer #4: 

This study is good I have suggested few revisions where so ever wording is not clear, It is looking at efficacy and safety of different anticoagulation regimens in plasma exchange and standards in US and Europe I am not sure why conclusion mentions that is not conclusive although study size for review is quiet enough

Thank you very much for the recognition of our work and the good words. However, the number of included studies is too limited (less than 10, so few that the publication bias assessment is not possible). Furthermore, no study had reported each examined outcome in this study. Therefore, we consider the results are insufficient to draw any conclusive statement at this stage. More solid evidence is expected from future studies. We have discussed this point as a limitation in the Discussion.

---

## [Decision Letter · Decision Letter 1]

20 Aug 2024

PONE-D-24-02893R1Comparison of Efficacy and Safety of Different Anticoagulation Regimens in Plasma Exchange: A Systematic Review and Meta-analysisPLOS ONE

Dear Dr. Ren,

Thank you for submitting your manuscript to PLOS ONE. After careful consideration, we feel that it has merit but does not fully meet PLOS ONE’s publication criteria as it currently stands. Therefore, we invite you to submit a revised version of the manuscript that addresses the points raised during the review process.

Thank you for revising the manuscript. Kindly comment and disclose about data gathering, reporting and storage if any that has not been included in the manuscript previously. My decision is justified on PLOS ONE’s publication criteria and not, for example, on novelty or perceived impact.

We look forward to receiving your revised manuscript.

Kind regards,

Vendhan Ramanujam, M.B.B.S, M.D.

Academic Editor

PLOS ONE

Journal Requirements:

Reviewers' comments:

Reviewer's Responses to Questions

**Comments to the Author**

1. If the authors have adequately addressed your comments raised in a previous round of review and you feel that this manuscript is now acceptable for publication, you may indicate that here to bypass the “Comments to the Author” section, enter your conflict of interest statement in the “Confidential to Editor” section, and submit your "Accept" recommendation.

Reviewer #1: All comments have been addressed

Reviewer #3: All comments have been addressed

2. Is the manuscript technically sound, and do the data support the conclusions?

Reviewer #1: Yes

Reviewer #3: No

3. Has the statistical analysis been performed appropriately and rigorously? 

Reviewer #1: Yes

Reviewer #3: Yes

4. Have the authors made all data underlying the findings in their manuscript fully available?

Reviewer #1: Yes

Reviewer #3: No

5. Is the manuscript presented in an intelligible fashion and written in standard English?

Reviewer #1: Yes

Reviewer #3: Yes

6. Review Comments to the Author

Reviewer #1: (No Response)

Reviewer #3: The manuscript "Comparison of Efficacy and Safety of Different Anticoagulation Regimens in Plasma Exchange: A Systematic Review and Meta-analysis" is technically sound, with appropriately performed and rigorously reported statistical analyses. The data support the conclusions, demonstrating that UFH is more effective than saline flushing and comparable to LMWH and RCA in anticoagulative efficacy. However, all underlying data is not made fully available, which is required by the PLOS Data Policy. They should ensure that all relevant data are provided within the manuscript, its supporting information, or deposited in a public repository. Additionally, given the complexity of the statistical methods, an additional review by a statistician is recommended to ensure accuracy. The manuscript addresses a critical clinical issue and offers valuable insights, making it suitable for highlighting on the PLOS ONE website to promote further research and discussion. There are no concerns about dual publication, research ethics, or publication ethics noted.

7. PLOS authors have the option to publish the peer review history of their article (what does this mean?). If published, this will include your full peer review and any attached files.

Reviewer #1: No

Reviewer #3: No

---

## [Author Response · Author response to Decision Letter 1]

24 Aug 2024

Reviewer #3: 

The manuscript "Comparison of Efficacy and Safety of Different Anticoagulation Regimens in Plasma Exchange: A Systematic Review and Meta-analysis" is technically sound, with appropriately performed and rigorously reported statistical analyses. The data support the conclusions, demonstrating that UFH is more effective than saline flushing and comparable to LMWH and RCA in anticoagulative efficacy. However, all underlying data is not made fully available, which is required by the PLOS Data Policy. They should ensure that all relevant data are provided within the manuscript, its supporting information, or deposited in a public repository. Additionally, given the complexity of the statistical methods, an additional review by a statistician is recommended to ensure accuracy. The manuscript addresses a critical clinical issue and offers valuable insights, making it suitable for highlighting on the PLOS ONE website to promote further research and discussion. There are no concerns about dual publication, research ethics, or publication ethics noted.

Thank you very much for the comment. The key data which we used for the data analysis are listed in Table 1 and Table 2 in the manuscript. To meet the requirement by PLOS Data Policy, we have now uploaded the full data set into the Dryad database (DOI: 10.5061/dryad.pvmcvdnvh) for readers’ interest. We have also made corresponding corrections in the “Data Availability” section in the manuscript which has been highlighted in yellow. We have also invited Mr. Qiang Li to review the data analysis process. Qiang is a senior statistician from The George Institute for Global Health in Sydney, Australia. He is an expert in medical statistic, has led statistical analysis in many international clinical studies, and is highly experienced. Based on Qiang’s assessment, we did not made change to the current results. Thank you very much for your help in this process.

---

## [Editor Report · Decision Letter 2]

23 Sep 2024

Comparison of Efficacy and Safety of Different Anticoagulation Regimens in Plasma Exchange: A Systematic Review and Meta-analysis

PONE-D-24-02893R2

Dear Dr. Ren,

We’re pleased to inform you that your manuscript has been judged scientifically suitable for publication and will be formally accepted for publication once it meets all outstanding technical requirements.

Kind regards,

Vendhan Ramanujam, M.B.B.S, M.D.

Academic Editor

PLOS ONE
---

## [Editor Report · Acceptance letter]

14 Oct 2024

PONE-D-24-02893R2 

PLOS ONE

Dear Dr. Ren, 

I'm pleased to inform you that your manuscript has been deemed suitable for publication in PLOS ONE. Congratulations! Your manuscript is now being handed over to our production team.

Kind regards, 

on behalf of

Dr. Vendhan Ramanujam 

Academic Editor

PLOS ONE